# Approximate Inference in Continuous Determinantal Point Processes

**Raja Hafiz Affandi**[1], **Emily B. Fox**[2], and **Ben Taskar**[2]

[1]University of Pennsylvania, `rajara@wharton.upenn.edu`
[2]University of Washington, {`ebfox@stat,taskar@cs`}`.washington.edu`

## Abstract

Determinantal point processes (DPPs) are random point processes well-suited for modeling repulsion. In machine learning, the focus of DPP-based models has been on diverse subset selection from a discrete and finite base set. This discrete setting admits an efficient sampling algorithm based on the eigendecomposition of the defining kernel matrix. Recently, there has been growing interest in using DPPs defined on continuous spaces. While the discrete-DPP sampler extends formally to the continuous case, computationally, the steps required are not tractable in general. In this paper, we present two efficient DPP sampling schemes that apply to a wide range of kernel functions: one based on low rank approximations via Nyström and random Fourier feature techniques and another based on Gibbs sampling. We demonstrate the utility of continuous DPPs in repulsive mixture modeling and synthesizing human poses spanning activity spaces.

## 1 Introduction

Samples from a determinantal point process (DPP) [15] are sets of points that tend to be spread out. More specifically, given $\Omega \subseteq \mathbb{R}^d$ and a positive semidefinite kernel function $L : \Omega \times \Omega \mapsto \mathbb{R}$, the probability density of a point configuration $A \subset \Omega$ under a DPP with kernel $L$ is given by

$$\mathcal{P}_L(A) \propto \det(L_A) \, , \tag{1}$$

where $L_A$ is the $|A| \times |A|$ matrix with entries $L(\mathbf{x}, \mathbf{y})$ for each $\mathbf{x}, \mathbf{y} \in A$. The tendency for repulsion is captured by the determinant since it depends on the volume spanned by the selected points in the associated Hilbert space of $L$. Intuitively, points similar according to $L$ or points that are nearly linearly dependent are less likely to be selected.

Building on the foundational work in [5] for the case where $\Omega$ is discrete and finite, DPPs have been used in machine learning as a model for subset selection in which diverse sets are preferred [2, 3, 9, 12, 13]. These methods build on the tractability of sampling based on the algorithm of Hough et al. [10], which relies on the eigendecomposition of the kernel matrix to recursively sample points based on their projections onto the subspace spanned by the selected eigenvectors.

Repulsive point processes, like *hard core processes* [7, 16], many based on thinned Poisson processes and Gibbs/Markov distributions, have a long history in the spatial statistics community, where considering continuous $\Omega$ is key. Many naturally occurring phenomena exhibit diversity—trees tend to grow in the least occupied space [17], ant hill locations are over-dispersed relative to uniform placement [4] and the spatial distribution of nerve fibers is indicative of neuropathy, with hard-core processes providing a critical tool [25]. Repulsive processes on continuous spaces have garnered interest in machine learning as well, especially relating to generative mixture modeling [18, 29].

The computationally attractive properties of DPPs make them appealing to consider in these applications. On the surface, it seems that the eigendecomposition and projection algorithm of [10] for discrete DPPs would naturally extend to the continuous case. While this is true in a formal sense as $L$

becomes an operator instead of a matrix, the key steps such as the eigendecomposition of the kernel and projection of points on subspaces spanned by *eigenfunctions* are computationally infeasible except in a few very limited cases where approximations can be made [14]. The absence of a tractable DPP sampling algorithm for general kernels in continuous spaces has hindered progress in developing DPP-based models for repulsion.

In this paper, we propose an efficient algorithm to sample from DPPs in continuous spaces using low-rank approximations of the kernel function. We investigate two such schemes: Nyström and random Fourier features. Our approach utilizes a *dual representation* of the DPP, a technique that has proven useful in the discrete $\Omega$ setting as well [11]. For $k$-DPPs, which only place positive probability on sets of cardinality $k$ [13], we also devise a Gibbs sampler that iteratively samples points in the $k$-set conditioned on all $k-1$ other points. The derivation relies on representing the conditional DPPs using the Schur complement of the kernel. Our methods allow us to handle a broad range of typical kernels and continuous subspaces, provided certain simple integrals of the kernel function can be computed efficiently. Decomposing our kernel into *quality* and *similarity* terms as in [13], this includes, but is not limited to, all cases where the (i) spectral density of the quality and (ii) characteristic function of the similarity kernel can be computed efficiently. Our methods scale well with dimension, in particular with complexity growing linearly in $d$.

In Sec. 2, we review sampling algorithms for discrete DPPs and the challenges associated with sampling from continuous DPPs. We then propose continuous DPP sampling algorithms based on low-rank kernel approximations in Sec. 3 and Gibbs sampling in Sec. 4. An empirical analysis of the two schemes is provided in Sec. 5. Finally, we apply our methods to repulsive mixture modeling and human pose synthesis in Sec. 6 and 7.

## 2  Sampling from a DPP

When $\Omega$ is discrete with cardinality $N$, an efficient algorithm for sampling from a DPP is given in [10]. The algorithm, which is detailed in the supplement, uses an eigendecomposition of the kernel matrix $L = \sum_{n=1}^{N} \lambda_n v_n v_n^{\top}$ and recursively samples points $\mathbf{x}_i$ as follows, resulting in a set $A \sim \text{DPP}(L)$ with $A = \{\mathbf{x}_i\}$:

Phase 1  Select eigenvector $v_n$ with probability $\frac{\lambda_n}{\lambda_n+1}$. Let $V$ be the selected eigenvectors ($k = |V|$).

Phase 2  For $i = 1, \ldots, k$, sample points $\mathbf{x}_i \in \Omega$ sequentially with probability based on the projection of $\mathbf{x}_i$ onto the subspace spanned by $V$. Once $\mathbf{x}_i$ is sampled, update $V$ by excluding the subspace spanned by the projection of $\mathbf{x}_i$ onto $V$.

When $\Omega$ is discrete, both steps are straightforward since the first phase involves eigendecomposing a kernel matrix and the second phase involves sampling from discrete probability distributions based on inner products between points and eigenvectors. Extending this algorithm to a continuous space was considered by [14], but for a very limited set of kernels $L$ and spaces $\Omega$. For general $L$ and $\Omega$, we face difficulties in both phases. Extending Phase 1 to a continuous space requires knowledge of the eigendecomposition of the kernel function. When $\Omega$ is a compact rectangle in $\mathbb{R}^d$, [14] suggest approximating the eigendecomposition using an orthonormal Fourier basis.

Even if we are able to obtain the eigendecomposition of the kernel function (either directly or via approximations as considered in [14] and Sec. 3), we still need to implement Phase 2 of the sampling algorithm. Whereas the discrete case only requires sampling from a discrete probability function, here we have to sample from a probability density. When $\Omega$ is compact, [14] suggest using a rejection sampler with a uniform proposal on $\Omega$. The authors note that the acceptance rate of this rejection sampler decreases with the number of points sampled, making the method inefficient in sampling large sets from a DPP. In most other cases, implementing Phase 2 even via rejection sampling is infeasible since the target density is in general non-standard with unknown normalization. Furthermore, a generic proposal distribution can yield extremely low acceptance rates.

In summary, current algorithms can sample approximately from a continuous DPP only for translation-invariant kernels defined on a compact space. In Sec. 3, we propose a sampling algorithm that allows us to sample approximately from DPPs for a wide range of kernels $L$ and spaces $\Omega$.

# 3 Sampling from a low-rank continuous DPP

Again considering $\Omega$ discrete with cardinality $N$, the sampling algorithm of Sec. 2 has complexity dominated by the eigendecomposition, $O(N^3)$. If the kernel matrix $L$ is low-rank, i.e. $L = B^\top B$, with $B$ a $D \times N$ matrix and $D \ll N$, [11] showed that the complexity of sampling can be reduced to $O(ND^2 + D^3)$. The basic idea is to exploit the fact that $L$ and the *dual kernel matrix* $C = BB^\top$, which is $D \times D$, share the same nonzero eigenvalues, and for each eigenvector $v_k$ of $L$, $Bv_k$ is the corresponding eigenvector of $C$. See the supplement for algorithmic details.

While the dependence on $N$ in the dual is sharply reduced, in continuous spaces, $N$ is infinite. In order to extend the algorithm, we must find efficient ways to compute $C$ for Phase 1 and manipulate eigenfunctions implicitly for the projections in Phase 2. Generically, consider sampling from a DPP on a continuous space $\Omega$ with kernel $L(\mathbf{x}, \mathbf{y}) = \sum_{n=1}^{\infty} \lambda_n \phi_n(\mathbf{x})\overline{\phi_n}(\mathbf{y})$, where $\lambda_n$ and $\phi_n(\mathbf{x})$ are eigenvalues and eigenfunctions, and $\overline{\phi_n}(\mathbf{y})$ is the complex conjugate of $\phi_n(\mathbf{y})$. Assume that we can approximate $L$ by a low-dimensional (generally complex-valued) mapping, $B(\mathbf{x}) : \Omega \mapsto \mathbb{C}^D$:

$$\tilde{L}(\mathbf{x}, \mathbf{y}) = B(\mathbf{x})^* B(\mathbf{y}) \text{ , where } B(\mathbf{x}) = [B_1(\mathbf{x}), \ldots, B_D(\mathbf{x})]^\top. \tag{2}$$

Here, $A^*$ denotes complex conjugate transpose of $A$. We consider two efficient low-rank approximation schemes in Sec. 3.1 and 3.2. Using such a low-rank representation, we propose an analog of the dual sampling algorithm for continuous spaces, described in Algorithm 1. A similar algorithm provides samples from a *k-DPP*, which only gives positive probability to sets of a fixed cardinality $k$ [13]. The only change required is to the for-loop in Phase 1 to select exactly $k$ eigenvectors using an efficient $O(Dk)$ recursion. See the supplement for details.

---

**Algorithm 1** Dual sampler for a low-rank continuous DPP

---

**Input:** $\tilde{L}(\mathbf{x}, \mathbf{y}) = B(\mathbf{x})^* B(\mathbf{y})$,      **PHASE 2**
       a rank-$D$ DPP kernel                $X \leftarrow \emptyset$
**PHASE 1**                                  **while** $|V| > 0$ **do**
Compute $C = \int_\Omega B(\mathbf{x})B(\mathbf{x})^* d\mathbf{x}$          Sample $\hat{\mathbf{x}}$ from $f(\mathbf{x}) = \frac{1}{|V|}\sum_{\mathbf{v} \in V} |\mathbf{v}^* B(\mathbf{x})|^2$
Compute eigendecomp. $C = \sum_{k=1}^{D} \lambda_k \mathbf{v}_k \mathbf{v}_k^*$       $X \leftarrow X \cup \{\hat{\mathbf{x}}\}$
$J \leftarrow \emptyset$                                     Let $\mathbf{v}_0$ be a vector in $V$ such that $\mathbf{v}_0^* B(\hat{\mathbf{x}}) \neq 0$
**for** $k = 1, \ldots, D$ **do**                  Update $V \leftarrow \{\mathbf{v} - \frac{\mathbf{v}^* B(\hat{\mathbf{x}})}{\mathbf{v}_0^* B(\hat{\mathbf{x}})}\mathbf{v}_0 \mid v \in V - \{v_0\}\}$
   $J \leftarrow J \cup \{k\}$ with probability $\frac{\lambda_k}{\lambda_k + 1}$      Orthonormalize $V$ w.r.t. $\langle \mathbf{v}_1, \mathbf{v}_2 \rangle = \mathbf{v}_1^* C \mathbf{v}_2$
$V \leftarrow \{\frac{v_k}{\sqrt{v_k^* C v_k}}\}_{k \in J}$                      **Output:** $X$

---

In this dual view, we still have the same two-phase structure, and must address two key challenges:

Phase 1   Assuming a low-rank kernel function decomposition as in Eq. (2), we need to able to compute the dual kernel matrix, given by an integral:

$$C = \int_\Omega B(\mathbf{x})B(\mathbf{x})^* d\mathbf{x} . \tag{3}$$

Phase 2   In general, sampling directly from the density $f(\mathbf{x})$ is difficult; instead, we can compute the cumulative distribution function (CDF) and sample $\mathbf{x}$ using the inverse CDF method [21]:

$$F(\hat{\mathbf{x}} = (\hat{x}_1, \ldots, \hat{x}_d)) = \prod_{l=1}^{d} \int_{-\infty}^{\hat{x}_l} f(\mathbf{x}) 1_{\{x_l \in \Omega\}} dx_l. \tag{4}$$

Assuming (i) the kernel function $\tilde{L}$ is finite-rank and (ii) the terms $C$ and $f(\mathbf{x})$ are computable, Algorithm 1 provides exact samples from a DPP with kernel $\tilde{L}$. In what follows, approximations only arise from approximating general kernels $L$ with low-rank kernels $\tilde{L}$. If given a finite-rank kernel $L$ to begin with, the sampling procedure is exact.

One could imagine approximating $L$ as in Eq. (2) by simply truncating the eigendecomposition (either directly or using numerical approximations). However, this simple approximation for known decompositions does not necessarily yield a tractable sampler, because the products of eigenfunctions required in Eq. (3) might not be efficiently integrable. For our approximation algorithm to work, not only do we need methods that approximate the kernel function well, but also that enable us to solve Eq. (3) and (4) directly for many different kernel functions. We consider two such approaches that enable an efficient sampler for a wide range of kernels: Nyström and random Fourier features.

### 3.1 Sampling from RFF-approximated DPP

Random Fourier features (RFF) [19] is an approach for approximating shift-invariant kernels, $k(\mathbf{x}, \mathbf{y}) = k(\mathbf{x} - \mathbf{y})$, using randomly selected frequencies. The frequencies are sampled independently from the Fourier transform of the kernel function, $\boldsymbol{\omega}_j \sim \mathcal{F}(k(\mathbf{x} - \mathbf{y}))$, and letting:

$$\tilde{k}(\mathbf{x} - \mathbf{y}) = \frac{1}{D} \sum_{j=1}^{D} \exp\{i\boldsymbol{\omega}_j^\top (\mathbf{x} - \mathbf{y})\}, \quad \mathbf{x}, \mathbf{y} \in \Omega. \tag{5}$$

To apply RFFs, we factor $L$ into a quality function $q$ and similarity kernel $k$ (i.e., $q(\mathbf{x}) = \sqrt{L(\mathbf{x}, \mathbf{x})}$):

$$L(\mathbf{x}, \mathbf{y}) = q(\mathbf{x})k(\mathbf{x}, \mathbf{y})q(\mathbf{y}), \qquad \mathbf{x}, \mathbf{y} \in \Omega \text{ where } k(\mathbf{x}, \mathbf{x}) = 1. \tag{6}$$

The RFF approximation can be applied to cases where the similarity function has a known characteristic function, e.g., Gaussian, Laplacian and Cauchy. Using Eq. (5), we can approximate the similarity kernel function to obtain a low-rank kernel and dual matrix:

$$\tilde{L}_{RFF}(\mathbf{x}, \mathbf{y}) = \frac{1}{D} \sum_{j=1}^{D} q(\mathbf{x}) \exp\{i\boldsymbol{\omega}_j^\top (\mathbf{x} - \mathbf{y})\}q(\mathbf{y}), \ C_{jk}^{RFF} = \frac{1}{D} \int_\Omega q^2(\mathbf{x}) \exp\{i(\boldsymbol{\omega}_j - \boldsymbol{\omega}_k)^\top \mathbf{x}\}d\mathbf{x}.$$

The CDF of the sampling distribution $f(\mathbf{x})$ in Algorithm 1 is given by:

$$F_{RFF}(\hat{\mathbf{x}}) = \frac{1}{|V|} \sum_{\mathbf{v} \in V} \sum_{j=1}^{D} \sum_{k=1}^{D} v_j v_k^* \prod_{l=1}^{d} \int_{-\infty}^{\hat{x}_l} q^2(\mathbf{x}) \exp\{i(\boldsymbol{\omega}_j - \boldsymbol{\omega}_k)^\top \mathbf{x}\}1_{\{x_l \in \Omega\}} dx_l. \tag{7}$$

where $v_j$ denotes the $j$th element of vector $\mathbf{v}$. Note that equations $C^{RFF}$ and $F_{RFF}$ can be computed for many different combinations of $\Omega$ and $q(\mathbf{x})$. In fact, this method works for any combination of (i) translation-invariant similarity kernel $k$ with known characteristic function and (ii) quality function $q$ with known spectral density. The resulting kernel $L$ need not be translation invariant. In the supplement, we illustrate this method by considering a common and important example where $\Omega = \mathbb{R}^d$, $q(\mathbf{x})$ is Gaussian, and $k(\mathbf{x}, \mathbf{y})$ is any kernel with known Fourier transform.

### 3.2 Sampling from a Nyström-approximated DPP

Another approach to kernel approximation is the Nyström method [27]. In particular, given $\mathbf{z}_1, \dots, \mathbf{z}_D$ *landmarks* sampled from $\Omega$, we can approximate the kernel function and dual matrix as,

$$\tilde{L}_{Nys}(\mathbf{x}, \mathbf{y}) = \sum_{j=1}^{D} \sum_{k=1}^{D} W_{jk}^2 L(\mathbf{x}, \mathbf{z}_j)L(\mathbf{z}_k, \mathbf{y}), \ C_{jk}^{Nys} = \sum_{n=1}^{D} \sum_{m=1}^{D} W_{jn} W_{mk} \int_\Omega L(\mathbf{z}_n, \mathbf{x})L(\mathbf{x}, \mathbf{z}_m)d\mathbf{x},$$

where $W_{jk} = L(\mathbf{z}_j, \mathbf{z}_k)^{-1/2}$. Denoting $\mathbf{w}_j(\mathbf{v}) = \sum_{n=1}^{D} W_{jn} v_n$, the CDF of $f(\mathbf{x})$ in Alg. 1 is:

$$F_{Nys}(\hat{\mathbf{x}}) = \frac{1}{|V|} \sum_{\mathbf{v} \in V} \sum_{j=1}^{D} \sum_{k=1}^{D} \mathbf{w}_j(\mathbf{v})\mathbf{w}_k(\mathbf{v}) \prod_{l=1}^{d} \int_{-\infty}^{\hat{x}_l} L(\mathbf{x}, \mathbf{z}_j)L(\mathbf{z}_k, \mathbf{x})1_{\{x_l \in \Omega\}} dx_l. \tag{8}$$

As with the RFF case, we consider a decomposition $L(\mathbf{x}, \mathbf{y}) = q(\mathbf{x})k(\mathbf{x}, \mathbf{y})q(\mathbf{y})$. Here, there are no translation-invariant requirements, even for the similarity kernel $k$. In the supplement, we provide the important example where $\Omega = \mathbb{R}^d$ and both $q(\mathbf{x})$ and $k(\mathbf{x}, \mathbf{y})$ are Gaussians and also when $k(\mathbf{x}, \mathbf{y})$ is polynomial, a case that cannot be handled by RFF since it is not translationally invariant.

## 4 Gibbs sampling

For $k$-DPPs, we can consider a Gibbs sampling scheme. In the supplement, we derive that the full conditional for the inclusion of point $\mathbf{x}_k$ given the inclusion of the $k-1$ other points is a 1-DPP with a modified kernel, which we know how to sample from. Let the kernel function be represented as before: $L(\mathbf{x}, \mathbf{y}) = q(\mathbf{x})k(\mathbf{x}, \mathbf{y})q(\mathbf{y})$. Denoting $J^{\setminus k} = \{\mathbf{x}_j\}_{j \neq k}$ and $M^{\setminus k} = L_{J^{\setminus k}}^{-1}$ the full conditional can be simplified using Schur's determinantal equality [22]:

$$p(\mathbf{x}_k | \{\mathbf{x}_j\}_{j \neq k}) \propto L(\mathbf{x}_k, \mathbf{x}_k) - \sum_{i,j \neq k} M_{ij}^{\setminus k} L(\mathbf{x}_i, \mathbf{x}_k)L(\mathbf{x}_j, \mathbf{x}_k). \tag{9}$$

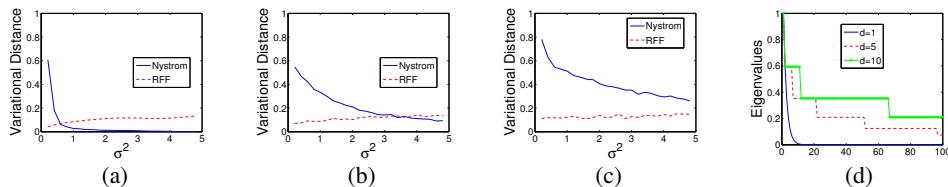

Figure 1: Estimates of total variational distance for Nyström and RFF approximation methods to a DPP with Gaussian quality and similarity with covariances $\Gamma = \text{diag}(\rho^2, \ldots, \rho^2)$ and $\Sigma = \text{diag}(\sigma^2, \ldots, \sigma^2)$, respectively. (a)-(c) For dimensions $d$=1, 5 and 10, each plot considers $\rho^2 = 1$ and varies $\sigma^2$. (d) Eigenvalues for the Gaussian kernels with $\sigma^2 = \rho^2 = 1$ and varying dimension $d$.

In general, sampling directly from this full conditional is difficult. However, for a wide range of kernel functions, including those which can be handled by the Nyström approximation in Sec. 3.2, the CDF can be computed analytically and $\mathbf{x}_k$ can be sampled using the inverse CDF method:

$$F(\hat{\mathbf{x}}_l | \{\mathbf{x}_j\}_{j \neq k}) = \frac{\int_{-\infty}^{\hat{\mathbf{x}}_l} L(\mathbf{x}_l, \mathbf{x}_l) - \sum_{i,j \neq k} M_{ij}^{\setminus k} L(\mathbf{x}_i, \mathbf{x}_l) L(\mathbf{x}_j, \mathbf{x}_l) 1_{\{\mathbf{x}_l \in \Omega\}} d\mathbf{x}_l}{\int_{\Omega} L(\mathbf{x}, \mathbf{x}) - \sum_{i,j \neq k} M_{ij}^{\setminus k} L(\mathbf{x}_i, \mathbf{x}) L(\mathbf{x}_j, \mathbf{x}) d\mathbf{x}} \quad (10)$$

In the supplement, we illustrate this method by considering the case where $\Omega = \mathbb{R}^d$ and $q(\mathbf{x})$ and $k(\mathbf{x}, \mathbf{y})$ are Gaussians. We use this same Schur complement scheme for sampling from the full conditionals in the mixture model application of Sec. 6. A key advantage of this scheme for several types of kernels is that the complexity of sampling scales linearly with the number of dimensions $d$ making it suitable in handling high-dimensional spaces.

As with any Gibbs sampling scheme, the mixing rate is dependent on the correlations between variables. In cases where the kernel introduces low repulsion we expect the Gibbs sampler to mix well, while in a high repulsion setting the sampler can mix slowly due to the strong dependencies between points and fact that we are only doing one-point-at-a-time moves. We explore the dependence of convergence on repulsion strength in the supplementary materials. Regardless, this sampler provides a nice tool in the $k$-DPP setting. Asymptotically, theory suggests that we get *exact* (though correlated) samples from the $k$-DPP. To extend this approach to standard DPPs, we can first sample $k$ (this assumes knowledge of the eigenvalues of $L$) and then apply the above method to get a sample. This is fairly inefficient if many samples are needed. A more involved but potentially efficient approach is to consider a birth-death sampling scheme where the size of the set can grow/shrink by 1 at every step.

## 5 Empirical analysis

To evaluate the performance of the RFF and Nyström approximations, we compute the total variational distance $\|\mathcal{P}_L - \mathcal{P}_{\tilde{L}}\|_1 = \frac{1}{2} \sum_X |\mathcal{P}_L(X) - \mathcal{P}_{\tilde{L}}(X)|$, where $\mathcal{P}_L(X)$ denotes the probability of set $X$ under a DPP with kernel $L$, as given by Eq. (1). We restrict our analysis to the case where the quality function and similarity kernel are Gaussians with isotropic covariances $\Gamma = \text{diag}(\rho^2, \ldots, \rho^2)$ and $\Sigma = \text{diag}(\sigma^2, \ldots, \sigma^2)$, respectively, enabling our analysis based on the easily computed eigenvalues [8]. We also focus on sampling from $k$-DPPs for which the size of the set $X$ is always $k$. Details are in the supplement.

Fig. 1 displays estimates of the total variational distance for the RFF and Nyström approximations when $\rho^2 = 1$, varying $\sigma^2$ (the repulsion strength) and the dimension $d$. Note that the RFF method performs slightly worse as $\sigma^2$ increases and is rather invariant to $d$ while the Nyström method performs much better for increasing $\sigma^2$ but worse for increasing $d$.

While this phenomenon seems perplexing at first, a study of the eigenvalues of the Gaussian kernel across dimensions sheds light on the rationale (see Fig. 1). Note that for fixed $\sigma^2$ and $\rho^2$, the decay of eigenvalues is slower in higher dimensions. It has been previously demonstrated that the Nyström method performs favorably in kernel learning tasks compared to RFF in cases where there is a large eigengap in the kernel matrix [28]. The plot of the eigenvalues seems to indicate the same phenomenon here. Furthermore, this result is consistent with the comparison of RFF to Nyström in approximating DPPs in the discrete $\Omega$ case provided in [3].

This behavior can also be explained by looking at the theory behind these two approximations. For the RFF, while the kernel approximation is guaranteed to be an unbiased estimate of the true kernel element-wise, the variance is fairly high [19]. In our case, we note that the RFF estimates of minors are biased because of non-linearity in matrix entries, overestimating probabilities for point

configurations that are more spread out, which leads to samples that are overly-dispersed. For the Nyström method, on the other hand, the quality of the approximation depends on how well the landmarks cover $\Omega$. In our experiments the landmarks are sampled i.i.d. from $q(\mathbf{x})$. When either the similarity bandwidth $\sigma^2$ is small or the dimension $d$ is high, the effective distance between points increases, thereby decreasing the accuracy of the approximation. Theoretical bounds for the Nyström DPP approximation in the case when $\Omega$ is finite are provided in [3]. We believe the same result holds for continuous $\Omega$ by extending the eigenvalues and spectral norm of the kernel matrix to operator eigenvalues and operator norms, respectively.

In summary, for moderate values of $\sigma^2$ it is generally good to use the Nyström approximation for low-dimensional settings and RFF for high-dimensional settings.

## 6 Repulsive priors for mixture models

Mixture models are used in a wide range of applications from clustering to density estimation. A common issue with such models, especially in density estimation tasks, is the introduction of redundant, overlapping components that increase the complexity and reduce interpretability of the resulting model. This phenomenon is especially prominent when the number of samples is small. In a Bayesian setting, a common fix to this problem is to consider a sparse Dirichlet prior on the mixture weights, which penalizes the addition of non-zero-weight components. However, such approaches run the risk of inaccuracies in the parameter estimates [18]. Instead, [18] show that sampling the location parameters using repulsive priors leads to better separated clusters while maintaining the accuracy of the density estimate. They propose a class of repulsive priors that rely on explicitly defining a distance metric and the manner in which small distances are penalized. The resulting posterior computations can be fairly complex.

The theoretical properties of DPPs make them an appealing choice as a repulsive prior. In fact, [29] considered using DPPs as repulsive priors in latent variable models. However, in the absence of a feasible continuous DPP sampling algorithm, their method was restricted to performing MAP inference. Here we propose a fully generative probabilistic mixture model using a DPP prior for the location parameters, with a $K$-component model using a $K$-DPP.

In the common case of mixtures of Gaussians (MoG), our posterior computations can be performed using Gibbs sampling with nearly the same simplicity of the standard case where the location parameters $\mu_k$ are assumed to be i.i.d.. In particular, with the exception of updating the location parameters $\{\mu_1, \ldots, \mu_K\}$, our sampling steps are identical to standard MoG Gibbs updates in the uncollapsed setting. For the location parameters, instead of sampling each $\mu_k$ independently from its conditional posterior, our full conditional depends upon the other locations $\mu_{\setminus k}$ as well. Details are in the supplement, where we show that this full conditional has an interpretation as a single draw from a tilted 1-DPP. As such, we can employ the Gibbs sampling scheme of Sec. 4.

We assess the clustering and density estimation performance of the DPP-based model on both synthetic and real datasets. In each case, we run 10,000 Gibbs iterations, discard 5,000 as burn-in and thin the chain by 10. Hyperparameter settings are in the supplement. We randomly permute the labels in each iteration to ensure balanced label switching. Draws are post-processed following the algorithm of [23] to address the label switching issue.

**Synthetic data**  To assess the role of the prior in a density estimation task, we generated a small sample of 100 observations from a mixture of two Gaussians. We consider two cases, the first with well-separated components and the second with poorly-separated components. We compare a mixture model with locations sampled i.i.d. (`IID`) to our DPP repulsive prior (`DPP`). In both cases, we set an upper bound of six mixture components. In Fig. 2, we see that both `IID` and `DPP` provide very similar density estimates. However, `IID` uses many large-mass components to describe the density. As a measure of simplicity of the resulting density description, we compute the average entropy of the posterior mixture membership distribution, which is a reasonable metric given the similarity of the overall densities. Lower entropy indicates a more concise representation in an information-theoretic sense. We also assess the accuracy of the density estimate by computing both (i) Hamming distance error relative to true cluster labels and (ii) held-out log-likelihood on 100 observations. The results are summarized in Table 1. We see that `DPP` results in (i) significantly lower entropy, (ii) lower overall clustering error, and (iii) statistically indistinguishable held-out log-likelihood. These results signify that we have a sparser representation with well-separated (interpretable) clusters while maintaining the accuracy of the density estimate.

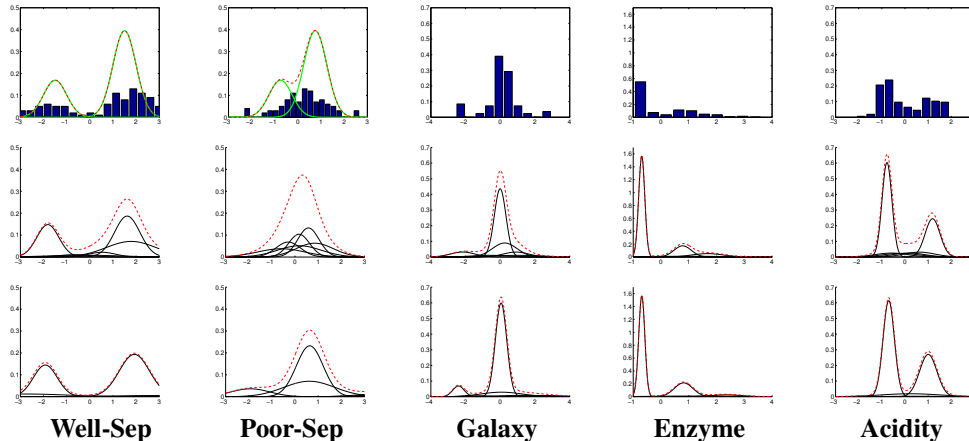

| Well-Sep | Poor-Sep | Galaxy | Enzyme | Acidity |
|---|---|---|---|---|

Figure 2: For each synthetic and real dataset: (top) histogram of data overlaid with actual Gaussian mixture generating the synthetic data, and posterior mean mixture model for (middle) `IID` and (bottom) `DPP`. Red dashed lines indicate resulting density estimate.

Table 1: For `IID` and `DPP` on synthetic datasets: mean (stdev) for mixture membership entropy, cluster assignment error rate and held-out log-likelihood of 100 observations under the posterior mean density estimate.

| DATASET | ENTROPY | | CLUSTERING ERROR | | HELDOUT LOG-LIKE. | |
|---|---|---|---|---|---|---|
| | IID | DPP | IID | DPP | IID | DPP |
| Well-separated | 1.11 (0.3) | 0.88 (0.2) | 0.19 (0.1) | 0.19 (0.1) | -169 (6) | -171(8) |
| Poorly-separated | 1.46 (0.2) | 0.92 (0.3) | 0.47 (0.1) | 0.39 (0.1) | -211(10) | -207(9) |

**Real data**   We also tested our DPP model on three real density estimation tasks considered in [20]: 82 measurements of velocity of galaxies diverging from our own (*galaxy*), acidity measurement of 155 lakes in Wisconsin (*acidity*), and the distribution of enzymatic activity in the blood of 245 individuals (*enzyme*). We once again judge the complexity of the density estimates using the posterior mixture membership entropy as a proxy. To assess the accuracy of the density estimates, we performed 5-fold cross validation to estimate the predictive held-out log-likelihood. As with the synthetic data, we find that `DPP` visually results in better separated clusters (Fig. 2). The `DPP` entropy measure is also significantly lower for data that are not well separated (*acidity* and *galaxy*) while the differences in predictive log-likelihood estimates are not statistically significant (Table 2).

Finally, we consider a classification task based on the *iris* dataset: 150 observations from three iris species with four length measurements. For this dataset, there has been significant debate on the optimal number of clusters. While there are three species in the data, it is known that two have very low separation. Based on loss minimization, [24, 26] concluded that the optimal number of clusters was two. Table 2 compares the classification error using `DPP` and `IID` when we assume for evaluation the real data has three or two classes (by collapsing two low-separation classes) , but consider a model with a maximum of six components. While both methods perform similarly for three classes, `DPP` has significantly lower classification error under the assumption of two classes, since `DPP` places large posterior mass on only two mixture components. This result hints at the possibility of using the DPP mixture model as a model selection method.

## 7   Generating diverse sample perturbations

We consider another possible application of continuous-space sampling. In many applications of inverse reinforcement learning or inverse optimal control, the learner is presented with control trajectories executed by an expert and tries to estimate a reward function that would approximately reproduce such policies [1]. In order to estimate the reward function, the learner needs to compare the rewards of a large set of trajectories (or all, if possible), which becomes intractable in high-dimensional spaces with complex non-linear dynamics. A typical approximation is to use a set of perturbed expert trajectories as a comparison set, where a good set of trajectories should cover as large a part of the space as possible.

Table 2: For `IID` and `DPP`, mean (stdev) of (*left*) mixture membership entropy and held-out log-likelihood for three density estimation tasks and (*right*) classification error under 2 vs. 2 of true classes for the *iris* data.

| DATA | ENTROPY | | HELDOUT LL. | |
|---|---|---|---|---|
| | IID | DPP | IID | DPP |
| Galaxy | 0.89 (0.2) | 0.74 (0.2) | -20(2) | -21(2) |
| Acidity | 1.32 (0.1) | 0.98 (0.1) | -49 (2) | -48(3) |
| Enzyme | 1.01 (0.1) | 0.96 (0.1) | -55(2) | -55(3) |

| DATA | CLASS ERROR | |
|---|---|---|
| | IID | DPP |
| Iris (3 cls) | 0.43 (0.02) | 0.43 (0.02) |
| Iris (2 cls) | 0.23 (0.03) | 0.15 (0.03) |

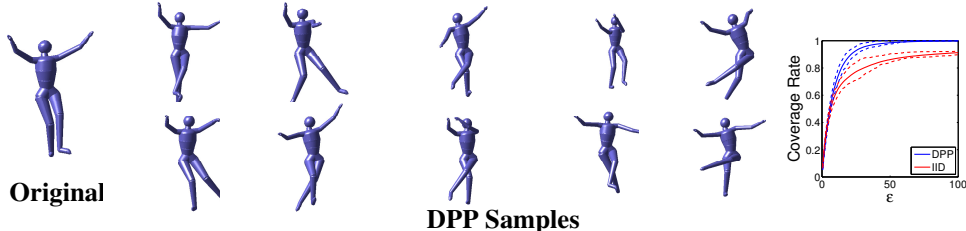

**Original**  **DPP Samples**

Figure 3: *Left:* Diverse set of human poses relative to an original pose by sampling from an RFF (top) and Nyström (bottom) approximations with kernel based on MoCap of the activity *dance*. *Right:* Fraction of data having a DPP/i.i.d. sample within an $\epsilon$ neighborhood.

We propose using DPPs to sample a large-coverage set of trajectories, in particular focusing on a human motion application where we assume a set of motion capture (MoCap) training data taken from the CMU database [6]. Here, our dimension $d$ is 62, corresponding to a set of joint angle measurements. For a given activity, such as *dancing*, we aim to select a reference pose and synthesize a set of diverse, perturbed poses. To achieve this, we build a kernel with Gaussian quality and similarity using covariances estimated from the training data associated with the activity. The Gaussian quality is centered about the selected reference pose and we synthesize new poses by sampling from our continuous DPP using the low-rank approximation scheme. In Fig. 3, we show an example of such DPP-synthesized poses. For the activity *dance*, to quantitatively assess our performance in covering the activity space, we compute a *coverage rate* metric based on a random sample of 50 poses from a DPP. For each training MoCap frame, we compute whether the frame has a neighbor in the DPP sample within an $\epsilon$ neighborhood. We compare our coverage to that of i.i.d. sampling from a multivariate Gaussian chosen to have variance matching our DPP sample. Despite favoring the i.i.d. case by inflating the variance to match the diverse DPP sample, the DPP poses still provide better average coverage over 100 runs. See Fig. 3 (right) for an assessment of the coverage metric. A visualization of the samples is in the supplement. Note that the i.i.d. case requires on average $\epsilon = 253$ to cover all data whereas the DPP only requires $\epsilon = 82$. By $\epsilon = 40$, we cover over 90% of the data on average. Capturing the rare poses is extremely challenging with i.i.d. sampling, but the diversity encouraged by the DPP overcomes this issue.

## 8 Conclusion

Motivated by the recent successes of DPP-based subset modeling in finite-set applications and the growing interest in repulsive processes on continuous spaces, we considered methods by which continuous-DPP sampling can be straightforwardly and efficiently approximated for a wide range of kernels. Our low-rank approach harnessed approximations provided by Nyström and random Fourier feature methods and then utilized a continuous dual DPP representation. The resulting approximate sampler garners the same efficiencies that led to the success of the DPP in the discrete case. One can use this method as a proposal distribution and correct for the approximations via Metropolis-Hastings, for example. For $k$-DPPs, we devised an exact Gibbs sampler that utilized the Schur complement representation. Finally, we demonstrated that continuous-DPP sampling is useful both for repulsive mixture modeling (which utilizes the Gibbs sampling scheme) and in synthesizing diverse human poses (which we demonstrated with the low-rank approximation method). As we saw in the MoCap example, we can handle high-dimensional spaces $d$, with our computations scaling just linearly with $d$. We believe this work opens up opportunities to use DPPs as parts of many models.

**Acknowledgements:** RHA and EBF were supported in part by AFOSR Grant FA9550-12-1-0453 and DARPA Grant FA9550-12-1-0406 negotiated by AFOSR. BT was partially supported by NSF CA-REER Grant 1054215 and by STARnet, a Semiconductor Research Corporation program sponsored by MARCO and DARPA.

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
