[Supplementary Material · supplementCR.pdf]

# Supplementary Material:
# Approximate Inference in Continuous Determinantal Processes

**Raja Hafiz Affandi**[1], **Emily B. Fox**[2], and **Ben Taskar**[2]

[1]University of Pennsylvania, `rajara@wharton.upenn.edu`
[2]University of Washington, {`ebfox@stat`,`taskar@cs`}`.washington.edu`

## Abstract

We provide further details for the NIPS 2013 submission "Approximate Inference in Continuous Determinantal Processes". First, we elaborate upon the existing DPP samplers for the discrete and finite $\Omega$ case. We then provide a list of standard cases when our (approximate) DPP sampling scheme can be performed. We derive the low-rank approximation and Gibbs sampling schemes for a few standard cases along with the details of empirical analysis of the low-rank approximations. For our mixture of Gaussian example application, we detail the model specification and Gibbs sampler and contrast with a standard (non-repulsive) mixture model. Finally, we provide additional details on the settings used in our experiments and present some additional figures of results.

## A  DPP, $k$-DPP, and dual DPP sampling

For $\Omega$ discrete and finite with cardinality $N$, we provide the algorithms for sampling from DPPs, $k$-DPPs, and DPPs via the dual representation in Algorithms 1, 2, 3. In the $k$-DPP sampler, $e_i$ denotes the $i$th elementary symmetric polynomial. For $\Omega$ continuous, we provide the continuous $k$-DPP dual sampler in Algorithm 4. Note that the only difference relative to the DPP dual sampler is in the for loop of Phase 1. The revision exactly parallels the story for the discrete $\Omega$ case.

---
**Algorithm 1** DPP-Sample(L)

---

**Input:** kernel matrix $L$ of rank $D$
**PHASE 1**
$\{(\boldsymbol{v}_n, \lambda_n)\}_{n=1}^{D} \leftarrow$ eigendecomposition of $L$
$J \leftarrow \emptyset$
**for** $n = 1, \ldots, D$ **do**
    $J \leftarrow J \cup \{n\}$ with prob. $\frac{\lambda_n}{\lambda_n + 1}$
$V \leftarrow \{\boldsymbol{v}_n\}_{n \in J}$

**PHASE 2**
$Y \leftarrow \emptyset$
**while** $|V| > 0$ **do**
    Select $i$ from $\Omega$ with $\Pr(i) = \frac{1}{|V|} \sum_{\boldsymbol{v} \in V} (\boldsymbol{v}^\top e_i)^2$
    $Y \leftarrow Y \cup \{i\}$
    $V \leftarrow V_{\perp e_i}$, an orthonormal basis for the subspace of
    V orthogonal to $e_i$
**Output:** $Y$

---

---
**Algorithm 2** $k$-DPP-Sample(L)
---
**Input:** kernel matrix $L$ of rank $D$, size $k$
**PHASE 1**
$\{(v_n, \lambda_n)\}_{n=1}^D \leftarrow$ eigendecomposition of $L$
$J \leftarrow \emptyset$
**for** $n = D, \dots, 1$ **do**
  **if** $u \sim U[0,1] < \lambda_n \frac{e_{k-1}^{n-1}}{e_k^n}$ **then**                       **PHASE 2** {same as Algorithm 1}
    $J \leftarrow J \cup \{n\}$
    $k \leftarrow k - 1$
    **if** $k = 0$ **then**
      **break**
$V \leftarrow \{\boldsymbol{v}_n\}_{n \in J}$
---

---
**Algorithm 3** Dual-DPP-Sample(B)
---
**Input:** $B \in \mathbb{C}^{D \times N}$ such that $L = B^* B$.
**PHASE 1**
$C \leftarrow BB^*$
$\{(\hat{\boldsymbol{v}}_n, \lambda_n)\}_{n=1}^D \leftarrow$ eigendecompistion of $C$
$J \leftarrow \emptyset$
**for** $n = 1, \dots, D$ **do**
  $J \leftarrow J \cup \{n\}$ with prob. $\frac{\lambda_n}{\lambda_n + 1}$
$\hat{V} \leftarrow \left\{ \frac{\hat{\boldsymbol{v}}_n}{\sqrt{\hat{\boldsymbol{v}}^* C \hat{\boldsymbol{v}}}} \right\}_{n \in J}$

**PHASE 2**
$Y \leftarrow \emptyset$
**while** $|\hat{V}| > 0$ **do**
  Select $i$ from $\Omega$ with $\Pr(i) = \frac{1}{|\hat{V}|} \sum_{\hat{\boldsymbol{v}} \in \hat{V}} (\hat{\boldsymbol{v}}^* B_i)^2$
  $Y \leftarrow Y \cup \{i\}$
  Let $\hat{\boldsymbol{v}}_0$ be a vector in $\hat{V}$ with $B_i^* \hat{\boldsymbol{v}}_0 \neq 0$
  Update $\hat{V} \leftarrow \left\{ \hat{\boldsymbol{v}} - \frac{\hat{\boldsymbol{v}}^* B_i}{\hat{\boldsymbol{v}}_0^* B_i} \hat{\boldsymbol{v}}_0 \mid \hat{\boldsymbol{v}} \in \hat{V} - \{\hat{\boldsymbol{v}}_0\} \right\}$
  Orthonormalize $\hat{V}$ w.r.t. $\langle \hat{\boldsymbol{v}}_1, \hat{\boldsymbol{v}}_2 \rangle = \hat{\boldsymbol{v}}_1^* C \hat{\boldsymbol{v}}_2$
**Output:** $Y$
---

---
**Algorithm 4** Dual sampler for a low-rank continuous $k$- DPP
---
**Input:** $\tilde{L}(\mathbf{x}, \mathbf{y}) = B(\mathbf{x})^* B(\mathbf{y})$,
      a rank-$D$ DPP kernel
**PHASE 1**
Compute $C = \int_\Omega B(\mathbf{x}) B(\mathbf{x})^* d\mathbf{x}$
$\{(\mathbf{v}_n, \lambda_n)\}_{n=1}^D \leftarrow$ eigendecomposition of $C$
**for** $n = D, \dots, 1$ **do**
  **if** $u \sim U[0,1] < \lambda_n \frac{e_{k-1}^{n-1}}{e_k^n}$ **then**
    $J \leftarrow J \cup \{n\}$
    $k \leftarrow k - 1$
    **if** $k = 0$ **then**
      **break**
$V \leftarrow \{ \frac{v_k}{\sqrt{v_k^* C v_k}} \}_{k \in J}$

**PHASE 2**
$X \leftarrow \emptyset$
**while** $|V| > 0$ **do**
  Sample $\hat{\mathbf{x}}$ from density $f(\mathbf{x}) = \frac{1}{|V|} \sum_{\mathbf{v} \in V} |\mathbf{v}^* B(\mathbf{x})|^2$
  $X \leftarrow X \cup \{\hat{\mathbf{x}}\}$
  Let $\mathbf{v}_0$ be a vector in $V$ such that $\mathbf{v}_0^* B(\hat{\mathbf{x}}) \neq 0$
  Update $V \leftarrow \{ \mathbf{v} - \frac{\mathbf{v}^* B(\hat{\mathbf{x}})}{\mathbf{v}_0^* B(\hat{\mathbf{x}})} \mathbf{v}_0 \mid v \in V - \{v_0\} \}$
  Orthonormalize $V$ w.r.t. $\langle \mathbf{v}_1, \mathbf{v}_2 \rangle = \mathbf{v}_1^* C \mathbf{v}_2$
**Output:** $X$
---

# B  Derivation of the Gibbs sampling scheme

For a $k$-DPP, the probability of choosing a specific $k$ point configuration is given by

$$p(\{\mathbf{x}_j\}_{j=1}^k) \propto \det(L_{\{\mathbf{x}_j\}_{j=1}^k}). \tag{1}$$

Denoting $J^{\backslash k} = \{\mathbf{x}_j\}_{j \neq k}$ and $M^{\backslash k} = L_{J^{\backslash k}}^{-1}$, the Schur's determinantal identity formula yields

$$\det(L_{\{\mathbf{x}_j\}_{j=1}^k}) = \det(L_{J^{\backslash k}}) \left( L(\mathbf{x}_k, \mathbf{x}_k) - \sum_{i,j \neq k} M_{ij}^{\backslash k} L(\mathbf{x}_i, \mathbf{x}_k) L(\mathbf{x}_j, \mathbf{x}_k) \right). \tag{2}$$

Conditioning on the inclusion of the other $k-1$ points, and suppressing constants not dependent on $\mathbf{x}_k$ we can now write the conditional distribution as

$$p(\mathbf{x}_k|\{\mathbf{x}_j\}_{j\neq k}) \propto L(\mathbf{x}_k,\mathbf{x}_k) - \sum_{i,j\neq k} M_{ij}^{\backslash k} L(\mathbf{x}_i,\mathbf{x}_k)L(\mathbf{x}_j,\mathbf{x}_k), \tag{3}$$

Normalizing and integrating this density yields a full conditional CDF given by

$$F(\hat{\mathbf{x}}_l|\{\mathbf{x}_j\}_{j\neq k}) = \frac{\int_{-\infty}^{\hat{\mathbf{x}}_l} L(\mathbf{x}_l,\mathbf{x}_l) - \sum_{i,j\neq k} M_{ij}^{\backslash k} L(\mathbf{x}_i,\mathbf{x}_l)L(\mathbf{x}_j,\mathbf{x}_l)1_{\{\mathbf{x}_l\in\Omega\}} d\mathbf{x}_l}{\int_{\Omega} L(\mathbf{x},\mathbf{x}) - \sum_{i,j\neq k} M_{ij}^{\backslash k} L(\mathbf{x}_i,\mathbf{x})L(\mathbf{x}_j,\mathbf{x}) d\mathbf{x}}. \tag{4}$$

## C  Overview of analytically tractable kernel types under RFF or Nyström

Sampling from a DPP with kernel $L$ using Algorithm 1 of the main paper requires that (i) we can compute a low-rank decomposition $\tilde{L}$ of $L$ and (ii) the terms $C$ and $f(\mathbf{x})$ are computable. In the main paper, we consider a decomposition of $L(\mathbf{x},\mathbf{y}) = q(\mathbf{x})k(\mathbf{x},\mathbf{y})q(\mathbf{y})$ where $q(\mathbf{x})$ is a quality function and $k(\mathbf{x},\mathbf{y})$ a similarity kernel. We then use either random Fourier features (RFF) or the Nyström method to approximate $L$ with $\tilde{L}$. In general, we can consider RFF approximations whenever the spectral density of $q(\mathbf{x})$ and characteristic function of $k(\mathbf{x},\mathbf{y})$ are known. For Nyström, the statement is not quite as clear. Instead, we provide a list of standard choices and their associated feasibilities for DPP sampling in Table 1. The list is by no means exhaustive, but is simply to provide some insight. We also elaborate upon some standard kernels in the following sections.

Table 1: Examination of the feasibility of DPP sampling using Nyström and RFF approximations for a few standard examples of quality functions $q$ and similarity kernels $k$.

| $q(x)$ | $k(x,y)$ | Method | |
|---|---|---|---|
| Gaussian, Laplacian | Gaussian, Laplacian | Nyström | ✓ |
| | | RFF | ✓ |
| | | Gibbs | ✓ |
| Gaussian, Laplacian | Cauchy | Nyström | ? |
| | | RFF | ✓ |
| | | Gibbs | ? |
| Cauchy | Gaussian, Laplacian | Nyström | ? |
| | | RFF | ✓ |
| | | Gibbs | ? |
| Cauchy | Cauchy | Nyström | ? |
| | | RFF | ✓ |
| | | Gibbs | ? |
| Gaussian, Laplacian | Linear, Polynomial | Nyström | ✓ |
| | | RFF | X |
| | | Gibbs | ✓ |

**Example: Sampling from RFF-approximated DPP with Gaussian quality**

Assuming $q(\mathbf{x}) = \exp\left\{-\frac{1}{2}(\mathbf{x}-\mathbf{a})^\top \Gamma^{-1}(\mathbf{x}-\mathbf{a})\right\}$ and $k(\mathbf{x},\mathbf{y}) = k(\mathbf{x}-\mathbf{y})$ is given by a translation-invariant kernel with known characteristic function. We start by sampling $\boldsymbol{\omega}_1,\ldots,\boldsymbol{\omega}_D \sim \mathcal{F}(k(\mathbf{x}-\mathbf{y}))$. Note, for example, that the Fourier transform of a Gaussian kernel is a Gaussian while that of the Laplacian is Cauchy and vice versa. The approximated kernel is given by

$$\tilde{L}_{RFF} = q(\mathbf{x})\left[\frac{1}{D}\sum_{j=1}^{D}\exp i\boldsymbol{\omega}_j{}^\top(\mathbf{x}-\mathbf{y})\right]q(\mathbf{y}) \ \ \text{where} \ \ q(\mathbf{x}) = \exp\left\{-\frac{1}{2}(\mathbf{x}-\mathbf{a})^\top \Gamma^{-1}(\mathbf{x}-\mathbf{a})\right\}. \tag{5}$$

The elements of the dual matrix $C^{RFF}$ are then given by

$$C_{jk}^{RFF} = \frac{1}{D} \int_{\mathbb{R}^d} \exp\{-(\mathbf{x} - \mathbf{a})^\top \Gamma^{-1}(\mathbf{x} - \mathbf{a}) + i(\boldsymbol{\omega}_j - \boldsymbol{\omega}_k)^\top \mathbf{x}\} d\mathbf{x}. \tag{6}$$

Letting $R\Delta R^\top$ be the spectral decomposition of $\Gamma^{-1}$ with $\Delta = \text{diag}(\frac{1}{\delta_1^2}, \ldots, \frac{1}{\delta_D^2})$, $\tilde{\boldsymbol{\omega}}_j = R^\top \boldsymbol{\omega}_j$, $\tilde{\mathbf{a}} = R^\top \mathbf{a}$ and $\mathbf{y} = R^\top \mathbf{x}$, one can straightforwardly derive:

$$C_{jk}^{RFF} = \frac{1}{D} \prod_{l=1}^{d} \left[ \sqrt{\pi \delta_l^2} \exp\left\{ -\frac{\delta_l^2(\tilde{\omega}_{jl} - \tilde{\omega}_{jk})^2}{4} \right\} + i\tilde{a}_l(\tilde{\omega}_{jl} - \tilde{\omega}_{jk}) \right]. \tag{7}$$

Likewise,

$$F_{RFF}(\mathbf{y}) = \frac{1}{D|V|} \sum_{\mathbf{v} \in V} \sum_{j=1}^{D} \sum_{k=1}^{D} \mathbf{v}^{(j)} \mathbf{v}^{(k)*} \prod_{l=1}^{d} g(\tilde{\omega}_{jl}, \tilde{\omega}_{kl}, \tilde{a}_l, \delta_l, y_l), \tag{8}$$

where

$$g(\tilde{\omega}_{jl}, \tilde{\omega}_{kl}, \tilde{\mathbf{a}}_l, \delta_l, y_l) = \frac{1}{2}\sqrt{\pi\delta_l^2} \exp\left\{ -\frac{\delta_l^2(\tilde{\omega}_{jl} - \tilde{\omega}_{kl})^2}{4} \right\} + i\tilde{a}_l\left( \tilde{\omega}_{jl} - \tilde{\omega}_{kl})(1 - \text{erf}\left( \frac{i\sqrt{\delta_l^2}(\tilde{\omega}_{jl} - \tilde{\omega}_{kl})}{2} - \frac{y_l - \tilde{a}_l}{2\sqrt{\delta_l^2}} \right) \right).$$

Once samples $\mathbf{y}$ are obtained, we transform back into our original coordinate system by letting $\mathbf{x} = R\mathbf{y}$.

**Example: Sampling from Nyström-approximated DPP with Gaussian quality and similarity**

Assuming $q(\mathbf{x}) = \exp\left\{ -\frac{1}{2}(\mathbf{x} - \mathbf{a})^\top \Gamma^{-1}(\mathbf{x} - \mathbf{a}) \right\}$ and $k(\mathbf{x}, \mathbf{y}) = \exp\left\{ -\frac{1}{2}(\mathbf{x} - \mathbf{y})^\top \Sigma^{-1}(\mathbf{x} - \mathbf{y}) \right\}$, the approximated kernel is given by

$$\tilde{L}_{Nys}(\mathbf{x}, \mathbf{y}) = \sum_{j=1}^{D} \sum_{k=1}^{D} W_{jk}^2 q(\mathbf{x}) q(\mathbf{z}_j) \exp\left\{ -\frac{1}{2}(\mathbf{x} - \mathbf{z}_j)^\top \Sigma^{-1}(\mathbf{x} - \mathbf{z}_j) - \frac{1}{2}(\mathbf{y} - \mathbf{z}_k)^\top \Sigma^{-1}(\mathbf{y} - \mathbf{z}_k) \right\} q(\mathbf{z}_k) q(\mathbf{y}). \tag{9}$$

Let $\Sigma^{-1} = Q\Lambda Q^\top$ with $\Lambda = \text{diag}(\frac{1}{\sigma_1^2}, \ldots, \frac{1}{\sigma_D^2})$, $\Gamma^{-1} = R\Delta R^\top$ with $\Delta = \text{diag}(\frac{1}{\delta_1^2}, \ldots, \frac{1}{\delta_D^2})$ and $(\Sigma^{-1} + \Gamma^{-1}) = T\Theta T^\top$ with $\Theta = \text{diag}(\frac{1}{\theta_1^2}, \ldots, \frac{1}{\theta_D^2})$. Furthermore, let $\tilde{\mathbf{z}}_j = T^\top(\Gamma^{-1} + \Sigma^{-1})\Sigma^{-1}\mathbf{z}_j$, $\tilde{\mathbf{a}} = T^\top(\Gamma^{-1} + \Sigma^{-1})\Gamma^{-1}\mathbf{a}$ and $\mathbf{y} = T^\top \mathbf{x}$. Then, the elements of the dual matrix $C^{Nys}$ are then given by

$$C_{jk}^{Nys} = \sum_{m-1}^{D} \sum_{n=1}^{D} W_{jn} W_{mk} A_{mn} \prod_{l=1}^{d} \sqrt{\pi \theta_l^2}. \tag{10}$$

where

$$\begin{aligned} A_{mn} = \exp\Big\{ &-\frac{1}{2}(\mathbf{z}_n - \mathbf{a})^\top \Gamma^{-1}(\mathbf{z}_n - \mathbf{a}) - \frac{1}{2}(\mathbf{z}_m - \mathbf{a})^\top \Gamma^{-1}(\mathbf{z}_m - \mathbf{a}) - \frac{1}{2}\mathbf{z}_m^\top \Sigma^{-1}\mathbf{z}_m - \frac{1}{2}\mathbf{z}_n^\top \Sigma^{-1}\mathbf{z}_n \\ &+ (\Gamma^{-1}\mathbf{a} + \Sigma^{-1}\frac{(\mathbf{z}_m + \mathbf{z}_n)}{2})^\top (\Sigma^{-1} + \Gamma^{-1})^{-1}(\Gamma^{-1}\mathbf{a} + \Sigma^{-1}\frac{(\mathbf{z}_m + \mathbf{z}_n)}{2}) - \mathbf{a}^\top \Gamma^{-1}\mathbf{a} \Big\}. \end{aligned}$$

Finally, the CDF of $f(\mathbf{y})$ is given by

$$F_{Nys}(\mathbf{y}) = \frac{1}{|V|} \sum_{\mathbf{v} \in V} \sum_{j,k=1}^{D} \mathbf{w}_j(\mathbf{v}) \mathbf{w}_k(\mathbf{v}) A_{jk} \prod_{l=1}^{d} \frac{\sqrt{\pi \theta_l^2}}{2} \left[ 1 - \text{erf}\left( \frac{2\tilde{a}_l + \tilde{z}_{jl} + \tilde{z}_{kl} - 2y_l}{2\sqrt{\theta_l^2}} \right) \right]. \tag{11}$$

Once samples $\mathbf{y}$ are obtained, we transform back to our original coordinate system by letting $\mathbf{x} = T\mathbf{y}$.

**Example: Sampling from Nyström-approximated DPP with Gaussian quality and polynomial similarity**

For simplicity of exposition, we consider a linear similarity kernel and $d = 1$, although the result can straightforwardly be extended to higher order polynomials and dimensions $d$. Assuming $q(x) = \exp\left\{-\frac{x^2}{2\rho^2}\right\}$ and $k(x, y) = xy$, the approximated kernel is given by

$$\tilde{L}_{Nys}(x, y) = \sum_{j=1}^{D}\sum_{k=1}^{D} W_{jk}^2 \exp\left\{-\frac{(x^2 + z_j^2 + z_k^2 + y^2)}{2\rho^2}\right\}(xz_j)(yz_k). \tag{12}$$

The elements of the dual matrix $C^{Nys}$ are then given by

$$C_{jk}^{Nys} = \sum_{m-1}^{D}\sum_{n=1}^{D} W_{jn}W_{mk}\frac{z_m z_n}{2}\exp\{-\frac{z_m^2 + z_n^2}{2\rho^2}\}\sqrt{\pi}\rho^3. \tag{13}$$

The CDF is given by

$$F_{Nys}(y) = \frac{1}{|V|}\sum_{\mathbf{v}\in V}\sum_{j,k=1}^{D} \mathbf{w}_j(\mathbf{v})\mathbf{w}_k(\mathbf{v})\frac{z_j z_k}{2}\exp\{-\frac{z_j^2 + z_k^2}{2\rho^2}\}\left[\frac{\sqrt{\pi}\rho^3}{4}\left[\text{erf}\left(\frac{y}{\sqrt{r}}\right) + 1\right] - 2ye^{-\frac{y^2}{\rho^2}}\right]. \tag{14}$$

**Example: Gibbs sampling with Gaussian quality and similarity**

For generic kernels $L(\mathbf{x}, \mathbf{y}) = q(\mathbf{x})k(\mathbf{x}, \mathbf{y})q(\mathbf{y})$, we recall that the CDF of $\mathbf{x}_k$ given $\{\mathbf{x}_j\}_{j\neq k}$ for a $k$-DPP is given by

$$F(\hat{\mathbf{x}}_k|\{\mathbf{x}_j\}_{j\neq k}) = \frac{\int_{-\infty}^{\hat{\mathbf{x}}_k} q(\mathbf{x}_k)^2(1 - \sum_{i,j\neq k} M_{ij}q(\mathbf{x}_i)q(\mathbf{x}_j)k(\mathbf{x}_k, \mathbf{x}_i)k(\mathbf{x}_j, \mathbf{x}_k))1_{\{\mathbf{x}_k\in\Omega\}}d\mathbf{x}_k}{\int_{\Omega} q(\mathbf{x})^2(1 - \sum_{i,j\neq k} M_{ij}q(\mathbf{x}_i)q(\mathbf{x}_j)k(\mathbf{x}, \mathbf{x}_i)k(\mathbf{x}_j, \mathbf{x}))d\mathbf{x}}. \tag{15}$$

Assuming $q(\mathbf{x}) = \exp\left\{-\frac{1}{2}(\mathbf{x} - \mathbf{a})^\top\Gamma^{-1}(\mathbf{x} - \mathbf{a})\right\}$ and $k(\mathbf{x}, \mathbf{y}) = \exp\left\{-\frac{1}{2}(\mathbf{x} - \mathbf{y})^\top\Sigma^{-1}(\mathbf{x} - \mathbf{y})\right\}$, the integrals above can be solved to yield

$$F(\hat{\mathbf{x}}_k|\{\mathbf{x}_j\}_{j\neq k}) = \frac{\prod_{l=1}^{d}\left[\frac{\sqrt{\pi\delta_l^2}}{2}\left[1 - \text{erf}\left(\frac{2\tilde{a}_l - 2x_{kl}}{2\sqrt{\delta_l^2}}\right)\right] - \sum_{i,j\neq k} M_{ij}A_{ij}\frac{\sqrt{\pi\theta_l^2}}{2}\left[1 - \text{erf}\left(\frac{2\tilde{a}_l + \tilde{z}_{il} + \tilde{z}_{jl} - 2x_{kl}}{2\sqrt{\theta_l^2}}\right)\right]\right]}{\prod_{l=1}^{d}\left[\sqrt{\pi\delta_l^2} - \sum_{i,j\neq k} W_{ij}A_{ij}\sqrt{\pi\theta_l^2}\right]}. \tag{16}$$

where $\tilde{\mathbf{a}}, \tilde{\mathbf{z}}, \delta_l, A_{ij}$ and $\theta_l$ are as given in the previous examples.

# D   Details of the empirical analysis

To evaluate the performance of the RFF and Nyström approximations, we compute the total variational distance

$$\|\mathcal{P}_L - \mathcal{P}_{\tilde{L}}\|_1 = \frac{1}{2}\sum_X |\mathcal{P}_L(X) - \mathcal{P}_{\tilde{L}}(X)|, \tag{17}$$

where $\mathcal{P}_L(X)$ denotes the probability of set $X$ under a DPP with kernel $L$, as given by Eq. (1). One can show that the normalized density is $\mathcal{P}_L(X) = \frac{\det(L_X)}{\prod_{n=1}^{\infty}(1+\lambda_n(L))}$, which requires the eigenvalues of the kernel $L$. Thus, we restrict our analysis to the case where the quality function and similarity kernel are Gaussians with isotropic covariances $\Gamma = \text{diag}(\rho^2, \ldots, \rho^2)$ and $\Sigma = \text{diag}(\sigma^2, \ldots, \sigma^2)$, respectively, since the eigenvalues of the kernel is easily computable in this setting [1]. In this case, letting $n = (n_1, \ldots, n_d)$ with $n_j \in \mathbb{Z}_+$, the eigenvalues (indexed by multi-index $n$) are given by:

$$\lambda_n = \prod_{j=1}^{d}\sqrt{\frac{\pi\rho^2}{\frac{c_1}{2} + c_2}}\left(\frac{1}{\frac{c_1}{c_2} + 1}\right)^{n_j-1} \qquad c_1 = (\beta^2 + 1) \qquad c_2 = \frac{\rho^2}{\sigma^2}. \tag{18}$$

where $\beta = (1 + \frac{2\rho^2}{\sigma^2})^{\frac{1}{4}}$. Since the eigenvalues are known in closed form, we can estimate the total variation distance by sampling sets $X$ from the approximated DPP and calculating the absolute difference between $\mathcal{P}_L(X)$ and $\mathcal{P}_{\tilde{L}}(X)$.

# E   Empirical analysis of Gibbs sampling

To assess the mixing rate of the Gibbs sampling scheme, we run the Gibbs sampler to sample points from a 1-dimensional 15-DPP with uniform quality and Gaussian similarity kernels in the space $\Omega = [-\frac{1}{2}, \frac{1}{2}]$. We perform this sampling under two values of repulsion parameter, $\sigma^2 = 0.01$ (high repulsion) and $\sigma^2 = 0.001$ (low repulsion). We run 100 Gibbs chains, each of length 3000, discard the first 1500 samples as burn-in and thin every 15 iterations which we call cycles. Each cycle represents a full resampling of the set, having cycled through the past 15 points. We compare the results to i.i.d. sampling of Nyström-approximated DPP as a baseline.

Figure 1 (a)-(b) shows a visualization of the 15 points of the 15-DPPs. Figure 1 (c)-(d) shows the plots of the Nyström-approximated DPP samples. As an ordered set, we see qualitatively that the locations of the points are highly correlated from cycle to cycle in the high repulsion Gibbs samples while less correlation is observed in the low-repulsion counterpart. In the Nyström approximated case, there are no correlations as the samples are generated i.i.d..

(a)                                                                 (b)

(c)                                                                 (d)

Figure 1: Visualization plots of location of 1-dimension DPP samples: (a)-(b) are samples from Gibbs scheme in low repulsion and high repulsion setting, respectively, (c)-(d) are i.i.d. samples from the Nyström-approximated DPP.

Quantitatively, we use two measures as a proxy to the mixing rate: the average movement of point from cycle to cycle and the effective sample size. The average movement, $m$, is simply defined as the average difference in distance

between points from one cycle to another averaged over the cycles:

$$m = \frac{1}{T-1}\frac{1}{k}\sum_{t=1}^{T-1}\sum_{i=1}^{k}(x_i^{t+1} - x_i^t)^2, \tag{19}$$

where $T$ is the length of the chain after burn-in and thinning, $k$ is the number of points and $x_i^t$ is the coordinate of point $x_i$ at cycle $t$. In our experiment, $T$ and $k$ are 100 and 15, respectively. When the Gibbs chain is mixing well, we expect the average movement to be high as this signals that the points are less correlated across cycles.

The effective sample size is a standard measure in assessing the mixing of a Gibbs chain. To compute this, we first compute the lag-$s$ autocorrelation function of each point in the sampled sets. We then average the autocorrelation function at lag-$s$ across the $k$ points and denote this quantity $\bar{\rho}_s$. The effective sample size is then given by: $\alpha T$, where

$$\alpha = \frac{1}{1 + 2\sum_{s=1}^{2\delta+1}\bar{\rho}_s}, \tag{20}$$

where $\delta$ is the smallest positive integer satisfying $\bar{\rho}_{2\delta} + \bar{\rho}_{2\delta+1} > 0$. In the case of i.i.d. samples, we expect $\alpha$ to be close to 1 while in cases where the mixing is bad, $\alpha$ will be much lower.

Table 2 shows the average values of $m$ and $\alpha$ for our Gibbs samples with i.i.d. Nyström-approximated DPP samples serving as a benchmark. We see that in the low repulsion setting, the Gibbs chain mixes well with values close to the benchmarks while for the Gibbs sampler in the high repulsion setting, the values of $m$ and $\alpha$ are much lower, indicating slow mixing.

|   | Gibbs High Repulsion | Gibbs Low Repulsion | Nyström High Repulsion | Nyström Low Repulsion |
|---|---|---|---|---|
| $m$ | 0.08 (0.07,0.08) | 0.1 (0.10,0.11) | 0.11 (0.1,0.11) | 0.11 (0.11,0.12) |
| $\alpha$ | 0.39 (0.31,0.45) | 0.92 (0.80,1) | 0.98 (0.82, 1) | 0.98 (0.90, 1) |

Table 2: The mean and 95% confidence interval for average movement, $m$ and the effective sample size coefficient, $\alpha$ for Gibbs samples and i.i.d. Nyström samples in high and low repulsion settings.

## F   Gibbs sampling for repulsive mixtures of Gaussians

Figure 2: Graphical models for mixtures of Gaussians using `IID` and `DPP` priors on the location parameters.

**Generative Model**   We consider a Bayesian mixture of Gaussians with either an independent normal (`IID`) or $K$-DPP (`DPP`) prior on the location parameters. In both cases, the $K$-component model with $N$ observations is specified as:

$$
\begin{aligned}
\pi \mid \alpha &\sim \text{Dir}(\alpha, \dots, \alpha) \\
\sigma_k^2 \mid a_\sigma, b_\sigma &\sim \text{IG}(a_\sigma, b_\sigma), \quad k = 1, \dots, K \\
\{\mu_1, \dots, \mu_K\} &\sim F \\
z_i \mid \pi &\sim \pi, \quad i = 1, \dots, N \\
y_i \mid \pi, \{\mu_k, \sigma_k^2\} &\sim N(\mu_{z_i}, \sigma_{z_i}^2), \quad i = 1, \dots, N.
\end{aligned}
\tag{21}
$$

---

**Algorithm 5** Mixture of Gaussians sampler

---

**Input:** Previous mixture weights $\pi$, emission parameters $\{\mu_k, \sigma_k\}^2$.
**for** $i = 1, \ldots, N$ **do**
    Sample cluster indicators $z_i \mid y_i, \{\mu_k, \sigma_k^2\}, \pi_k \propto \frac{1}{C_i} \sum_{k=1}^{K} \pi_k N(y_i; \mu_k, \sigma_k^2) \delta(z_i, k)$
Sample mixture weights $\pi \mid \{z_i\}, \alpha \sim \text{Dir}(\alpha + N_1, \ldots, \alpha + N_K)$
**for** $k = 1, \ldots, K$ **do**
    Sample scale parameters $\sigma_k^2 \mid \{y_i : z_i = k\}, \mu_k, a_\sigma, b_\sigma \sim \text{IG}\left(a_\sigma + \frac{N_k}{2}, b_\sigma + \frac{1}{2}\sum_{i:z_i=1}(y_i - \mu_k)^2\right)$
Sample location parameters $\{\mu_1, \ldots, \mu_K\} \mid \{y_i\}, \{z_i\}, \{\sigma_k^2\} \sim F_{post}$
**Output:** New mixture weights $\pi$, emission parameters $\{\mu_k, \sigma_k^2\}$.

---

Here, IG denotes the inverse gamma distribution and Dir a $K$-dimensional Dirichlet. For simplicity, we consider the univariate case here, though the multivariate case follows directly by considering an inverse Wishart prior in place of the inverse gamma and likewise modifying $F$ accordingly. Such a multivariate case is examined in the *iris* classification example in the main paper.

The difference between the models is in how the location parameters are specified. For the $\texttt{IID}$ case, we simply have:

$$\mu_k \mid \mu_0, \sigma_0^2 \sim N(\mu_0, \sigma_0^2) \tag{22}$$

For the $\texttt{DPP}$ case, we jointly sample:

$$\{\mu_1, \ldots, \mu_K\} \mid L \sim K\text{-DPP}(L). \tag{23}$$

We consider $L$ decomposed into Gaussian quality and similarity terms:

$$L(\mu_m, \mu_n) = q(\mu_m)k(\mu_m, \mu_n)q(\mu_n), \tag{24}$$

with

$$k(\mu_m, \mu_n) = \exp\left\{-\frac{(\mu_m - \mu_n)^2}{\gamma_0^2}\right\}, \quad q(\mu_m) = N(\mu_0, 2\sigma_0^2). \tag{25}$$

**Gibbs sampling**    For the uncollapsed setting, where mixture weights $\pi$ and emission parameters $\{\mu_k, \sigma_k^2\}$ are sampled, Algorithm 5 summarizes the Gibbs sampler for the finite mixture of Gaussians. We write the algorithm generically so that the overlap between $\texttt{IID}$ and $\texttt{DPP}$ is clear. In particular, the locations are sampled from $F_{post}$, which generically refers to the full conditional of the cluster means. For the $\texttt{IID}$ case, we sample i.i.d. for each $k$ from

$$\mu_k \mid \{y_i : z_i = k\}, \sigma_k^2, \mu_0, \sigma_0^2 \sim N\left(\hat{\mu}_k, \hat{\sigma}_k^2\right), \tag{26}$$

where $\hat{\mu}_k = \left(\frac{1}{\sigma_0^2} + \frac{N_k}{\sigma_k^2}\right)^{-1}\left(\frac{\mu_0}{\sigma_0^2} + \frac{1}{\sigma_k^2}\sum_{i:z_i=k} y_i\right)$ and $\hat{\sigma}_k^2 = \left(\frac{1}{\sigma_0^2} + \frac{N_k}{\sigma_k^2}\right)^{-1}$. Here, $N_k = |\{y_i : z_i = k\}|$, i.e., the cardinality of the set of observations assigned to cluster $k$.

For $\texttt{DPP}$, note that $p(\{\mu_j\}_{j=1}^k | \{y_i\}, \{z_i\}, \{\mu_k, \sigma_k^2\}) \propto \det(L_{\mu_1,\ldots,\mu_k}) \prod_{j=1}^k \prod_{i:z_i=j} N(y_i; \mu_j, \sigma_j^2)$. Unfortunately, this posterior distribution is not a $k$-DPP. However, fixing the rest of $k-1$ centroids, the full conditional of $\mu_k$ is (dropping constant terms that do not depend on $\mu_k$)

$$p(\mu_k | \{y_i\}, \{z_i\}, \{\mu_j, \sigma_j^2\}_{j\neq k}, \sigma_k^2) \propto \det(L_{\mu_1,\ldots,\mu_k}) \prod_{i:z_i=k} N(y_i; \mu_k, \sigma_k^2). \tag{27}$$

As before, we can use Schur's determinantal equality [2] to get

$$\det(L_{\mu_1,\ldots,\mu_k}) \propto L(\mu_k, \mu_k) - \sum_{i,j\neq k} M_{ij}^{\setminus k} L(\mu_i, \mu_k)L(\mu_j, \mu_k) \tag{28}$$

$$= q^2(\mu_k)\left(1 - \sum_{i,j\neq k} M_{ij}^{\setminus k} q(\mu_i)k(\mu_i, \mu_k)k(\mu_j, \mu_k)q(\mu_j)\right). \tag{29}$$

*Full conditional for μ_k*

*Data assigned to cluster k*

I.I.D.

DPP

*Other centroids*

Figure 3: Comparison between the full conditional for $\mu_k$ using the `IID` and `DPP` models at a given iteration $m$ of the sampler.

Combining the previous two equations, we get the full conditional

$$p(\mu_k|\{y_i\}, \{z_i\}, \{\mu_j, \sigma_j^2\}_{j\neq k}, \sigma_k^2) \propto q^2(\mu_k) \left(1 - \sum_{i,j\neq k} M_{ij}^{\backslash k} q(\mu_i) k(\mu_i, \mu_k) k(\mu_j, \mu_k) q(\mu_j)\right) \prod_{i:z_i=k} N(y_i; \mu_k, \sigma_k^2).$$

(30)

The CDF of the distribution above can be computed easily, since it only involves exponential quadratic forms. The inverse CDF method can then be used to obtain a sample from the above distribution. Note once again that $q^2(\mu_k) \prod_{i:z_i=k} N(y_i; \mu_k, \sigma_k^2)$ is defined to be exactly the same as the Gaussian distribution where $\mu_k$ would have been sampled from in the `IID` case. Thus the equation above gives a nice intuition on the conditional density of $\mu_k$ in the `DPP` setting: it is an exponentially tilted distribution in which $q^2(\mu_k) \prod_{i:z_i=k} N(y_i; \mu_k, \sigma_k^2)$ is corrected by a factor that depends on the location of the other centroids. In the case where all of the other centroids are far away from the cluster center $\hat{\mu}_k$, the correction factor is close to one and we would recover the density for the `IID` case.

To get a sense of why the `DPP` leads to more diverse cluster centers than `IID`, consider the full conditional for $\mu_k$ at some iteration $m$ of our sampler, as visualized in Fig. 3. We have some data points currently assigned to cluster $k$ via cluster indicators $z_i = k$. The `IID` model assumes that $\mu_k$ is independent of the other $\mu_j$'s whereas the `DPP` conditions on the other cluster centers leading to a conditional distribution for $\mu_k$ that puts more mass on uncovered regions. In subsequent iterations, the data that had been assigned to cluster $k$ but are not well covered by the sampled (and repulsed) $\mu_k$ will instead be assigned to one of the existing cluster centers that have mass near that data item. Such an alternative cluster exists, and is why $\mu_k$ was repulsed from that region, or will likely exist in future draws.

One attractive aspect of our `DPP` formulation is the fact that the sampling strategy maintains nearly the same simplicity as the standard `IID` sampler. This is in contrast, for example, to the repulsive mixture formulation of [3] which relied on slice sampling and draws from truncated normals, where the truncating region could only be computed in closed form for a restricted set of repulsive functions.

# G  Additional details on experiments

## G.1  Hyperparameter settings

For our mixture of Gaussian experiments, we used an inverse Wishart $\text{IW}(\nu, \Psi)$ with $\nu = d + 1$ and $\Psi = I$, which corresponds to $a_\sigma = 2$ and $b_\sigma = 1$ for the inverse Gamma in 1-dimension. Here, we use an inverse Wishart specification

Figure 4: (a)-(c) DPP (blue) and i.i.d. multivariate Gaussian (red) samples projected onto the top 4 principal components of the *dance* data.

such that $\Sigma \sim \text{IW}(\nu, \Psi)$ has mean $E[\Sigma] = \frac{\Psi}{\nu - d + 1}$. The Dirichlet hyperparameters were set to $\alpha = \frac{1}{3}$, just as in [3]. For the location hyperparameters, in the `IID` case we set $\mu_0 = 0$ and $\sigma_0^2 = 1$. In the `DPP` case, we use $\mu_0$ and $\sigma_0^2$ as in the `IID` case and set the repulsion parameter $\rho_0^2 = 1$.

For the MoCap experiment, we computed the covariance estimate from the training data, and set the similarity covariance parameter $\Sigma$ equal to this estimate. We then take the quality covariance parameter to be $\Gamma = \frac{1}{2}\Sigma$.

### G.2  Additional figures for MoCap experiments

In Fig. 4, we provide a visualization of poses sampled from the DPP relative to i.i.d. sampling of poses from a multivariate Gaussian. From these plots, we see how the sample of poses from the DPP covers a broader space, even when the covariance of the multivariate Gaussian is inflated to match that of the DPP. The reason for this broader coverage is the fact that the under the DPP, sampled poses repulse from regions already covered by other sampled poses.

Fig. 5 displays additional human poses that are drawn i.i.d. from a multivariate Gaussian, and compares to our DPP draws from both the RFF and Nyström approximations.

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

Original Pose

Poses synthesized from i.i.d. draws from a multivariate Gaussian

Poses synthesized from an RFF-approximated DPP

Poses synthesized from a Nyström-approximated DPP

Figure 5: Synthesizing perturbed human poses relative to an original pose by sampling (1) i.i.d. from a multivarite Gaussian versus (2) drawing a set from an RFF- or Nyström- approximated DPP with kernel based on MoCap data from the activity category. The Gaussian covariance is likewise formed from the activity data.