[Reviews · NeurIPS 2013]

Submitted by Assigned_Reviewer_5

This paper extends determinantal point process (DPP) sampling schemes
from discrete to continuous spaces by exploiting low-rank kernels.
Standard kernels are approximated using the Nystroem or
random Fourier features methods. The method is demonstrated on
repulsive mixture models and separated sampling.

The paper exploits a low rank kernel in order to derive an efficient
"dual sampler", as described in Alg 1. The authors show in which cases
of quality function and similarity kernel the necessary computations
can be carried out for the RFF and Nystroem approximations (Supp tab
1). This is a non-trivial piece of work.

In sec 5 the continuous DPP is used as a prior on the means
in a mixture model. However, note that the approximation
derived before is not actually used; as the authors say
"since we are sampling just one item from the DPP there is a closed
form and we do not rely on approximations." The results show that
DPPs do (as expected) have a repulsive effect, as evinced by
the mixture weight entropies.

Aside: surely to qualify as a repulsive mixture, the closeness of the
means should really be relative to their variances, but in supp mat p
4 the variances are independent of the means ...

In sec 6 a DPP is used to sample a diverse set of poses from a given
density. While the resulting poses are credibly diverse, I suspect
that visually similar results could have been obtained e.g. using hard
core processes (see p 1), or some scheme like farthest point
clustering (Gonzales, 1985).

Quality: As far as I can judge the material is technically correct.

Clarity: Generally clear.

As to Fig 1, I wanted more detail as to the defn of (15) -- for a
given set X we can evaluate this, but how is the sum defined and
evaluated?

Originality: as the authors make clear, this work extends earlier work
on low-rank approximations in discrete spaces. And as they mention the
repulsive prior construction has been used before [20,32].

Significance: it is not very clear about the applicability of the main
result. Sec 5 does not actually use it, and sec 6 is just a
visualization which does not require a precise sampler. Of course it
is possible that the result could be exploited in other interesting
ways.


Other:

* l 102. [11] are not the first to show that a matrix with the form B^T B
(D << N) has a fast eigendecomposition, while the form of the citation
implies it is. For example this result was used in
L. Sirovich and M. Kirby (1987). "Low-dimensional procedure for the
characterization of human faces". Journal of the Optical Society of
America A 4 (3): 519–524, but is surely
in any good advanced linear algebra text, e.g. Wilkinson?

* in eq 2 define the \bar{ \phi_n} notation (presumably as complex
conjugate).

* l 241-242 why not sigma^2 I instead of diag(sigma^2, \ldots, sigma^2),
and the same with rho^2?

* l 304 -- why use only 100 test observations? Reduce variance
by using many more!

Summary: This paper extends determinantal point process (DPP) sampling schemes
from discrete to continuous spaces by exploiting low-rank kernels.
Standard kernels are approximated using the Nystroem or
random Fourier features methods. This is a non-trivial piece of work.
The method is demonstrated on repulsive mixture models and separated sampling.


Submitted by Assigned_Reviewer_6

The goal of this paper is to sample from a continuous
determinantal point process, which until now has not been done
for general continuous kernels. The authors provide a general
`meta'-algorithm that works in principal for general kernels ---
though ease-of-derivation may vary from kernel to kernel.

Previous work dealt with translation invariant kernels in compact domains.
The random Fourier features method proposed here extends the
applicability of random Fourier features to general Euclidean spaces
(where the kernel is translation invariant). And the proposed Nystrom
method extends this even further to general kernels but relies
on good landmark selection (and generally performs better in low
dimensional problems).

While the RFF and Nystrom methods had been used in the discrete
version of the DPP, the main novelty of the current work is to
figure out how to do parts of the approximation `analytically' in some
special cases. In particular, explicit sampling algorithms are worked
out for Gaussian/Cauchy/Laplace kernel settings.

Overall, I quite enjoyed reading the paper as it was well motivated
and accessible. The NIPS margins have been violated, however, which made
the paper quite long. Otherwise, it makes for a nice contribution
to the DPP thread in machine learning and I recommend acceptance.

Minor comments/suggestions:
- I would have appreciated more detail in the derivation of the Gibbs
sampling step for the repulsive mixture. The authors seem to have
just written down a posterior which takes the form of a DPP. It
would be easier to follow if the prior and likelihood for the
`interesting' sampling step were written out explicitly.
- In Algorithm 1 in the appendix, the authors use Y and script-Y,
which are *not* the same, but somewhat confusing on a first read.
- It would be interesting to see a derivation of the Nystrom method
on a non-translation invariant kernel (to show off what it can
handle that the Fourier based features cannot).

Summary: This paper forms a nice extension to the exciting DPP (determinantal point processes) line of work in machine learning. It was also a pleasure to read. I hope it is accepted.

Submitted by Assigned_Reviewer_7

This paper describes a method for approximately sampling from certain
parameteric forms of a continuous-state determinantal point process. It
employs an eigen-decomposition (of the dual kernel matrix instead of the
kernel function) and depends on the kernel function having particular forms
and decompositions.

In the experimental results, the authors demonstrate that when coupled with
a mixture density model, the resulting models give better test performance
by promoting more diverse clusters.

The paper is presented well and the results are nice. The topic is of
interest to the ML community.

A few minor comments:

The notation for n (above equation 16) is unclear. Although I think I
understand the paper well, I am uncertain what n_j means.

The figure labels in the text do not align (for instance in Section 5,
Figure 5 is referred to).
Summary: The paper shows an interesting way of sampling from a continuous DPP. The results demonstrate clear utility. A good paper.
Author Feedback

Author rebuttal: We thank the reviewers for their thoughtful and helpful comments and we plan to incorporate these suggestions into the paper. We address major questions below.

R5:

Means/variances: Scaling the repulsion force by taking into account covariances is a good idea when the mixture components are not isotropic and can be potentially incorporated into our Gibbs sampler that iterates over means and variances.

Gibbs vs Joint sampling. We presented two samplers: Low-rank joint sampler for continuous DPPs and Gibbs coordinate-wise sampler suitable for mixture models and other models with DPP priors. For mixture models, conditioned on the data, the posterior over all the means is not a DPP, so our low-rank sampler cannot be used directly. However, using the same analytical integration approach, Gibbs coordinate-wise sampling over means is efficient. The low-rank sampler can of course be used in more general Metropolis samplers as a proposal. Having closed form formulas for many of the quantities (normalization, marginals) is very useful for constructing complex samplers, while hard core processes and the non-probabilistic diversity schemes are arguably harder to compose with other samplers.


The sum in Eqn.15/Fig.1: We use Monte Carlo sampling and an analytic expression for the normalizer for Gaussian quality/similarity k-DPP kernel to compute an estimate of the variational distance. We used a 1000 samples for each value of sigma, which was sufficient to accurately separate the errors of the two approximations.

R6:

Y/script Y notation in the appendix: this notation is used for discrete spaces in previous work.

More details on Gibbs derivations and translation-variant kernels (e.g. polynomial) will be included in supplemental material.

R7:

n=(n_1,..n_d) denotes a multi index used to refer to eigenvalues for kernels in higher dimensions.